# Inductive Synthesis of Finite-State Controllers for POMDPs

Roman Andriushchenko[1]     Milan Češka[1]     Sebastian Junges[2]     Joost-Pieter Katoen[3]

[1]Brno University of Technology, Brno, Czech Republic
[2]Radboud University, Nijmegen, The Netherlands
[3]RWTH Aachen University, Aachen, Germany

## Abstract

We present a novel learning framework to obtain finite-state controllers (FSCs) for partially observable Markov decision processes and illustrate its applicability for indefinite-horizon specifications. Our framework builds on oracle-guided inductive synthesis to explore a design space compactly representing available FSCs. The inductive synthesis approach consists of two stages: The outer stage determines the design space, i.e., the set of FSC candidates, while the inner stage efficiently explores the design space. This framework is easily generalisable and shows promising results when compared to existing approaches. Experiments indicate that our technique is (i) competitive to state-of-the-art belief-based approaches for indefinite-horizon properties, (ii) yields smaller FSCs than existing methods for several POMDP models, and (iii) naturally treats multi-objective specifications.

## 1 INTRODUCTION

Partially observable MDPs (POMDPs) model sequential decision making processes in which the agent only observes limited information about the current state of the system [Smallwood and Sondik, 1973, Kaelbling et al., 1998]. The key challenge in the analysis of POMDPs is to compute a policy satisfying some constraints, captured as a threshold on (discounted) reward or as a task description given in, e.g., a temporal logic. In full generality, policies need arbitrary memory to reflect the belief state of the agent. Point-based [Pineau et al., 2006, Spaan and Vlassis, 2005] and Monte Carlo methods [Silver and Veness, 2010] excel in finding such policies. Solving the generally undecidable policy learning problem profits from having complementary approaches in the portfolio. A natural alternative is to search for (small) finite-state controllers (FSCs) [Hansen, 1998].

Such controllers provide benefits in terms of explainability [Bonet et al., 2010, Wang and Niepert, 2019], resource-consumption [Grześ et al., 2013], and generalisability [Inala et al., 2020]. Recently, an automata learning framework has been proposed for synthesising permissive FSCs [Wu et al., 2021]. In this paper, we propose a novel approach—*inductive synthesis*—to find FSCs for POMDPs.

Inductive synthesis is a technique developed in the context of program synthesis, originally proposed by Church in the 1950's, the task to construct a program that provably satisfies a given formal specification. As developing a program (or in this context, a controller) from scratch is mostly infeasible, variants emerged, most notably syntax-guided synthesis [Alur et al., 2015, 2018] variations such as *sketching* [Solar-Lezama et al., 2006]. In sketching, the user provides a sketch that outlines a controller implementation, and a specification that constrains the controller's behaviour. The principal engine behind many instances of sketching is (oracle-guided) *inductive synthesis* [Jha and Seshia, 2017] and falls in a more general framework of learner-teacher frameworks. In a nutshell, this methodology suggests to heuristically *guess* candidate solutions, to *validate* them, and in case the solution is not satisfactory, *learn* in order to refine the search heuristic. The successful application of inductive synthesis has inspired numerous applications beyond classical programming, including recent works on sketching of probabilistic programs [Nori et al., 2015, Ceska et al., 2021, Andriushchenko et al., 2021b] and (variations of) programmatic reinforcement learning [Verma et al., 2018, Inala et al., 2020]. *This paper proposes inductive synthesis to search for FSCs in POMDPs.*

Our inductive synthesis framework works in two stages, see Fig. 1. Let us first discuss the outer stage. Here, a learner constructs a *design space* containing (finitely many) FSCs. A teacher provides the 'best' FSC within this design space, and potentially additional diagnostic information. The learner either accepts the FSC provided by the teacher as final result, or adapts the design space. Naturally, teachers will provide much better FSCs much faster whenever the design space

*Accepted for the 38th Conference on Uncertainty in Artificial Intelligence* (UAI 2022).

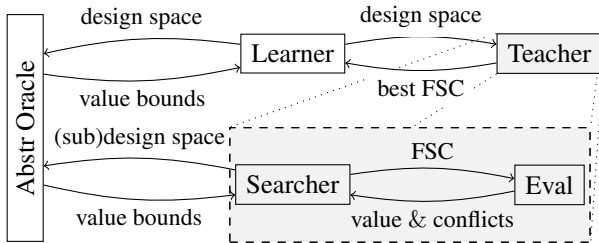

Figure 1: Nested inductive synthesis framework with an abstraction oracle. The framework takes a POMDP and a specification and finds an FSC that satisfies the specification.

for these FSCs is small. The key ingredient for the outer stage is thus to start with a small design space and to strategically modify this design space based on the obtained feedback from the teacher. A similar idea was proposed in [Kumar and Zilberstein, 2015], where the entropy of the observations is used as criterion for adding memory to the FSC. We use the FSC returned by the teacher together with the state-values induced by this FSC. Additionally, we use an abstraction oracle, see below.

The inner stage describes the internals of the teacher that determines the 'best' FSC within the design space. The teacher may naively use enumeration, but can also be realised using branch-and-bound [Grześ et al., 2013] or mixed-integer linear programming (MILP) [Amato et al., 2010, Kumar and Zilberstein, 2015]. We realise the teacher by (another) inductive synthesis loop. We search for an FSC by symbolically representing the design space as a propositional logic formula. The policy evaluation analyses a fixed policy w.r.t. the given specification (e.g., a reward function and a threshold). If the policy refutes the specification, the evaluation engine indicates the distance to satisfaction (e.g., the achieved value) as well as conflicts—critical parts of the FSC that suffice to violate the specification—that are used to prune the search design space [Ceska et al., 2021].

Both learning stages have access to an additional oracle that, inspired by Andriushchenko et al. [2021a], *over-approximates* the design space. This larger abstract design space can efficiently be analysed as the underlying problem solved by the abstraction oracle resembles the analysis of fully observable policies. The oracle yields constraints to what the best FSC within the original design space will possibly achieve. This information is an essential ingredient to guide the search in both stages.

The separate policy evaluation—a natural component in an inductive synthesis framework—brings some advantages. The policy evaluation (i.e., solving systems of linear equations) via dedicated algorithms is faster than letting an (MI)LP solver solve these equations [Dehnert et al., 2014]. This improves upon performance of MILP-based approaches (either primal [Winterer et al., 2020] or dual [Kumar and Zilberstein, 2015]) for FSC synthesis. Further-

more, as the policy is fixed, our framework provides an elegant alternative to existing approaches for constrained POMDPs [Poupart et al., 2015, Khonji et al., 2019] and multi-objective POMDPs [Soh and Demiris, 2011, Roijers et al., 2013, Wray and Zilberstein, 2015]. It additionally paves the way to learn robust FSCs for POMDPs with imprecise probabilities, similar to [Cubuktepe et al., 2021].

We instantiate our framework to learn *deterministic* FSCs, i.e., FSCs that do not use randomisation. Finding optimal deterministic FSCs is NP-complete whereas finding optimal randomised FSCs is ETR-complete[1] [Junges et al., 2018, 2021]. Algorithmically, finding randomised FSCs requires solving non-convex optimisation problems with thousands of variables. This often limits the guarantees on global (almost-)optimality that are practically feasible [Kumar and Zilberstein, 2015]. Deterministic FSCs are additionally beneficial in terms of reproducibility of their behaviour, which is useful for debugging. We use an evaluation framework that supports indefinite horizon queries, e.g., queries with a discount factor one. These queries generalize infinite horizon properties with a discount factor $<1$ and finite-horizon settings as used in Goal-POMDPs [Bonet and Geffner, 2009, Kolobov et al., 2011]. These queries naturally occur when using temporal logic specifications and are particularly adequate for safety-critical aspects.

The experimental evaluation shows the applicability of our approach on a wide range of benchmarks with promising results. Particularly, it significantly outperforms approaches based on MILP optimisation. We further compare it with the state-of-the-art belief-based approaches, namely, with recent works in formal verification on under-approximation for indefinite-horizon specifications [Norman et al., 2017, Bork et al., 2022]. Our inductive synthesis approach is highly competitive and for several POMDPs (having a moderate number of observations/actions and large/infinite belief-space), it is able to find small FSCs improving lower bounds of existing solutions.

## 2  PROBLEM STATEMENT

A *(discrete) distribution* over a finite set $X$ is a function $\mu \colon X \to [0,1]$ s.t. $\sum_x \mu(x) = 1$. The set $Distr(X)$ contains all distributions over $X$.

A *Markov decision process (MDP)* is a tuple $M = (S, s_0, Act, P)$ with a finite set $S$ of *states*, an initial state $s_0 \in S$, a finite set $Act$ of *actions*, and a *transition probability function* $P(s' \mid s, a)$ that gives the probability of evolving to $s'$ after taking action $a$ in $s$. A *Markov chain* (MC) is an MDP with $|Act| = 1$; its transition function is written as $P(s' \mid s)$. MDPs can additionally be equipped with a reward function $r(s, a)$. We do not use discount factors, see the paragraph on specifications below.

---

[1]The class ETR lies between NP and PSPACE.

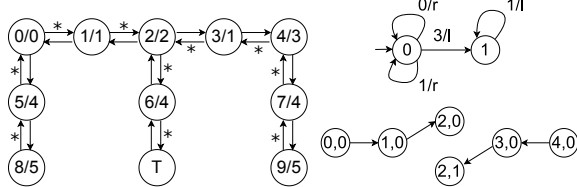

Figure 2: A simple maze problem (left), a part of a 2-FSC (right, top) and a part of the induced MC (right, bottom).

A *Partially Observable MDP (POMDP)* $\mathcal{M} = (S, s_0, Act, P, Z, O)$ extends MDP $M$ with a finite set $Z$ of *observations*, and a (deterministic) *observation function* [2] $O$ that returns for every state $s$ an observation $O(s) = z \in Z$. The observation $z \in Z$ is said to be *trivial* if there is only one state $s \in S$ with $O(s) = z$.

**Finite State Controllers (FSCs)** are automata that compactly represent policies. We call its states (memory) nodes to distinguish them from POMDP states. We also refer to an FSC with $k$ nodes as having $k$ memory. FSCs in the literature come in various styles, in particular either as Moore machines, with the output—the action it selects—determined by the node, or as Mealy machines, with the output determined by the taken transition [Amato et al., 2010]. In the context of sketching FSCs and their inductive exploration, it is convenient to describe FSCs as Mealy machines. Furthermore, we restrict ourselves to deterministic FSCs.

Formally, a *finite-state controller* (FSC) for a POMDP $\mathcal{M}$ is a tuple $F = (N, n_0, \gamma, \delta)$, where $N$ is a finite set of *nodes*, $n_0 \in N$ is the *initial node*, $\gamma(n, z)$ determines the action when the agent is in node $n$ and observes $z$, while $\delta$ updates the memory node to $\delta(n, z)$, when being in $n$ and observing $z$. For $|N| = k$, we call an FSC a $k$-FSC.

Imposing $k$-FSC $F$ onto POMDP $\mathcal{M}$ yields the *induced* Markov chain $\mathcal{M}^F = (S^F, (s_0, n_0), P^F)$ with $S^F = S \times N$ and using[3] $z = O(s)$:

$$P^F((s', n') \mid (s, n)) = P(s' \mid s, \gamma(n, z)) \cdot [\delta(n, z) = n'].$$

**Example 1.** As running example, we use a simple variant of the maze problem [Hauskrecht, 1997], where an agent tries to reach the state $s_T$, modelled by the POMDP $\mathcal{M}$ with $S = \{s_0, \ldots s_9, s_T\}$, $Act = \{u, d, l, r\}$, and $Z = \{z_0, \ldots, z_5\}$. The initial state is given by a uniform distribution over $S$. Fig. 2 (left) depicts $P$ and $O$ where state $s_x$ is labelled by $x/y$ with $x$ the state index and $y$ its observation, i.e., $O(s_x) = z_y$. The arrow direction from $x/y$ to $x'/y'$ represents the action; e.g., $\rightarrow$ corresponds to action $r$. The maze is slippery. An action is successful with probability 0.9; with 0.1, the agent does not move. Actions

without effect are omitted from the figure. Fig. 2 (right, top) illustrates a fragment of a 2-FSC where $\gamma(n_0, z_0) = \gamma(n_0, z_1) = r$ (for memory node $n_0$ and observations $z_0$ and $z_1$, action $r$ is chosen), $\gamma(n_0, z_3) = \gamma(n_1, z_1) = l$, $\delta(n_0, z_0) = \delta(n_0, z_1) = n_0$ (memory node $n_0$ is not changed for $z_0$ an $z_1$) and $\delta(n_0, z_3) = \delta(n_1, z_1) = n_1$. This FSC tries to resolve the inconsistency (formalised later) in the observation $z_1$, i.e., in $s_1$ the action $r$ is optimal but in $s_3$ action $l$ is optimal (w.r.t. reaching $s_T$). Fig. 2 (right, bottom) illustrates a fragment of the induced MC containing two copies of $s_2$.

**Specifications** contain two parts: a set of *constraints* given by quantitative properties and a single optimisation *objective*. Constraints are defined as indefinite-horizon reachability and expected reward properties, but our approach also supports more general probabilistic temporal logic properties [Baier and Katoen, 2008][4] Let target set $T \subseteq S$, *thresholds* $\lambda_1 \in [0, 1]$ and $\lambda_2 \in \mathbb{R}^+$ and $\bowtie \in \{\leq, \geq\}$. The POMDP $\mathcal{M}$ under FSC $F$ *satisfies* the constraint $P_{\bowtie \lambda_1}$ if the probability $\Pr^F$ of reaching $T$ in the induced MC $\mathcal{M}^F$ meets $\bowtie \lambda_1$. Similarly, the constraint $R_{\bowtie \lambda_2}$ is satisfied if the expected reward $R^F$ accumulated in MC $\mathcal{M}^F$ until reaching $T$ meets $\bowtie \lambda_2$. We call an FSC $F$ *admissible* (for $\mathcal{M}$), if $\mathcal{M}$ under $F$ satisfies the given (set of) constraint(s). Objectives either minimise or maximise reachability probabilities (as in goal-POMDPs) or (un)discounted expected reward properties, denoted as $P_*$ and $R_*$ respectively for $* \in \{\min, \max\}$. The probability or reward obtained by FSC $F$ on $\mathcal{M}$ is called the *value* of $F$. For conciseness, we assume throughout the paper that the specification contains a maximisation objective. Minimisation is analogously supported (but may require flipping bounds and inequalities).

**Problem statement.** We aim to construct an algorithm that: i) quickly finds a (small) admissible FSC $F$ and ii) incrementally improves $F$ w.r.t. the optimisation objective. We can view the algorithm as solving a sequence of decision problems, where the first decision problem is to find some admissible FSC $F_0$ and decision problem $i+1$ is to find an admissible FSC $F_{i+1}$ whose value improves upon the value of the previous FSC $F_i$.

## 3 INDUCTIVE EXPLORATION OF FSCS

This section presents the inner loop (see Fig. 1) in which we search among a given set of $k$-FSCs. Before we describe the ingredients, we formalise the representation of the set of $k$-FSCs. We then outline the two oracles that our search can use to prune the search space. A *hybrid strategy* [Andriushchenko et al., 2021a] combines the two oracles by switching based on perceived performance while communication between the oracles takes place.

---

[2] Observation functions resulting in a distribution over observations can be encoded by deterministic observation functions at the expense of a polynomial blow-up [Chatterjee et al., 2016].

[3] Iverson-brackets: $[x] = 1$ if predicate $x$ is true, 0 otherwise.

[4] These properties can describe the setting of goal-POMDPs, finite horizon reachability and rewards, and discounted rewards.

## 3.1 FAMILIES OF FSCS

A POMDP and a single FSC yield a single induced MC. A POMDP and a set of FSCs thus induces a set of MCs. The set of FSCs has additional structure which enables concisely describing the set of MCs. We first consider *full* FSCs where for each observation the same amount of memory is used and where there are no restrictions on the memory updates. We generalise this to a class of *reduced* FSCs that are more memory efficient.

**Definition 1.** A *family* of full $k$-FSCs is a tuple $\mathcal{F}_k = (N, n_0, K)$, where $N$ is a set consisting of $k$ nodes, $n_0 \in N$ is the initial node, $K = N \times Z$ is a finite set of parameters each with domain $V_{(n,z)} \subseteq Act \times N$.

From a family, one may obtain a $k$-FSC by choosing values for each parameter, effectively determining the action $\gamma(n, z)$ and the next node $\delta(n, z)$. Thus, each family describes a set of FSCs by varying the substitutions of the parameters. We often use $\mathcal{F}_k$ to denote such a set of $k$-FSCs. We remark that this set contains $\mathcal{O}((|Act||N|)^{(|N||Z|)})$ many FSCs. A POMDP $\mathcal{M}$ and a family $\mathcal{F}_k$ naturally induces the family of MCs $\mathcal{M}^{\mathcal{F}_k} = \{\mathcal{M}^F \mid F \in \mathcal{F}_k\}$.

**Example 2.** The family $\mathcal{F}_2$ of all 2-FSCs for our maze problem is given by $N = \{n_0, n_1\}$, $K = \{(n_i, z_j) \mid i \in \{0, 1\} \wedge j \in \{0, \dots, 5\}\}$, and $V_{(n,z)} = \{u, d, l, r\} \times \{n_0, n_1\}$ for all $(n, z) \in K$.

While FSCs have $k$ available memory nodes in conjunction with every observation, memory may only be required in some observation (see e.g., the running example). Therefore, we consider reduced FSCs given by a *memory restriction* $\mu : Z \to \mathbb{N}$, where $\mu(z)$ determines the number of memory nodes used in the observation $z$.

**Definition 2.** A *reduced family* $\mathcal{F}_\mu$ given by the memory model $\mu$ is a sub-family of $\mathcal{F}_k$ for $k = \max_{z \in Z}\{\mu(z)\}$ where $(n, z) \in K$ implies $n \leq \mu(z)$, and the domains $V_{(n,z)}$ are as in $\mathcal{F}_k$. If a memory update $\delta(n, z) = n'$ is invalid in the resulting observation $z'$ (i.e., $n' > \mu(z')$), then $\delta(n, z) = n_0$.

The reduced family for $k$-FSCs yields a significantly reduced number of parameters $\sum_{z \in Z}\{\mu(z)\}$. This is less than $k \cdot |Z|$ if $\mu(z) < k$ for most observations $z$. Indeed, in many experiments, we can use $\mu(z) = 1$ for all but a few observations. Such reduction has several key benefits: Foremost, the family of reduced FSCs induces a smaller design space, but it also yields FSCs where less memory is needed, which can be beneficial for their interpretability.

## 3.2 MDP-ABSTRACTION TEACHER

We introduce a teacher which, for a POMDP and a FSC-family, provides safe upper and lower bounds on the value of the FSCs in this family. Instead of considering the individual FSCs, the oracle considesr an abstraction (represented as a single MDP) of the set of induced MCs.

**Definition 3.** MDP $\mathcal{A}^{\mathcal{F}} = (S \times N, (s_0, n_0), Act^{\mathcal{F}}, P^{\mathcal{F}})$ is an *abstraction* of MC family $\mathcal{M}^{\mathcal{F}}$ with $Act^{\mathcal{F}} = Act \times N$ and $P^{\mathcal{F}}((s', n') \mid (s, n), (a, n')) = P(s' \mid s, a)$ if $(a, n') \in V_{(n, O(s))}$, and 0 otherwise.

For MDPs and our specifications, it suffices to consider deterministic memoryless policies, i.e., policy $\pi$ for MDP $\mathcal{A}^{\mathcal{F}}$ is a function $\pi : S \times N \to Act^{\mathcal{F}}$. It is *consistent* (w.r.t. the observations) if $O(s) = O(s')$ implies $\pi((s, n)) = \pi((s', n))$ for all $s, s' \in S, n \in N$. The set of consistent policies in $\mathcal{A}^{\mathcal{F}}$ corresponds to the policies for the family $\mathcal{F}$. The policy $\pi$ is *inconsistent* in a FSC-family parameter $(n, z) \in K$ if $\exists s, s' \in S : O(s) = O(s') = z \wedge \pi((s, n)) \neq \pi((s', n))$. It is inconsistent in observation $z \in Z$, if it is inconsistent in the parameter $(n, z) \in K$ for some $n \in N$.

**Example 3.** Assume we want to maximise the probability to reach $s_T$. The stars in Fig. 2 represent the optimal policy $\pi^*$ in MDP $\mathcal{A}^{\mathcal{F}}$ where $\mathcal{F}$ is set of all 1-FSCs for the maze problem. $\pi^*$ is inconsistent in the observations $z_1$ and $z_4$.

The analysis of MDP $\mathcal{A}^{\mathcal{F}}$ provides useful information about the family $\mathcal{F}$. For conciseness, consider the constraint $P_{\geq \lambda}$ bounding the reachability probability to states in $T$. We can compute the policy $\pi^*$ in $\mathcal{A}^{\mathcal{F}}$ that maximises this reachability probability. In particular, this policy achieves probability $\Pr^{\pi^*}$. If $\Pr^{\pi^*} < \lambda$, then all $F \in \mathcal{F}$ violate the constraint $P_{\geq \lambda}$ and $\mathcal{F}$ can be safely discarded. Otherwise, we check the consistency of policy $\pi^*$. If $\pi^*$ is consistent, it represents an FSC satisfying $P_{\geq \lambda}$. Similarly, a minimising policy may witness that the entire family $\mathcal{F}$ satisfies $P_{\geq \lambda}$. If analysing $\mathcal{A}^{\mathcal{F}}$ is inconclusive, we refine $\mathcal{F}$ by splitting, see below.

The optimisation objective is handled by iteratively updating a new (initially trivial) constraint representing the running value of the optimum so far. Once an admissible policy $\pi$ is found, we update the new constraint according to the objective value that $\pi$ achieves. Reasoning about multiple constraints works as follows. If the entire family $\mathcal{F}$ violates some constraint, $\mathcal{F}$ is discarded. If $\mathcal{F}$ satisfies the constraint, this constraint will neither be checked again for $\mathcal{F}$ nor for its subfamilies. Otherwise, if the analysis of $\mathcal{F}$ was inconclusive with respect to some constraint, $\mathcal{F}$ is refined.

Beyond pruning families, analysing $\mathcal{A}^{\mathcal{F}}$ provides state-vectors $ub$ and $lb$ such that $\forall s \in S$, $lb(s)$ and $ub(s)$ bound the probability to reach $T$ from $s$. These bounds are used in the inner and outer synthesis loop, as we will see below.[5]

---

[5]Furthermore, the state-vectors $ub$ and $lb$ allow *bootstrapping* the analysis of MDP $\mathcal{A}^{\mathcal{F}_i}$ where $\mathcal{F}_i$ is a subfamily of $\mathcal{F}$: This exploits the fact that $\mathcal{F}_i$ shares the structure of $\mathcal{F}$ while some actions for some states are removed.

**General refinement strategy** The refinement strategy is a key component in driving the exploration of the family $\mathcal{F}$. It decomposes $\mathcal{F}$ into sub-families by splitting the domain of selected parameters from $K$. In contrast to the general strategy used in program synthesis [Ceska et al., 2019], we leverage the specific topology of the FSC families.

The key idea is to examine the inconsistencies of the policy $\pi^*$ obtained for a given constraint. Assume $\pi^*$ is a maximising policy inconsistent in parameter $(n, z) \in K$. We estimate the *significance* of this inconsistency as the average variance (with respect to inconsistent actions) of $ub((n, s))$ with $O(s) = z$, where $ub((n, s))$ is weighted by the expected number of visits of the state $(n, s)$ in the MC induced by $\pi^*$ (if $\pi^*$ is minimising, we use $lb$). We refine $\mathcal{F}$ using the most significant inconsistent parameter $(n, z)$. Assume it has domain $V_{(n,z)}$ and $\pi^*$ selected options $v_1, \ldots, v_i$. We partition $V_{(n,z)}$ into $\{v_1\}, \ldots, \{v_i\}$ and $V_{(n,z)} \setminus \{v_1, \ldots, v_i\}$, and create $i+1$ corresponding subfamilies. This removes the inconsistency of $(n, z)$ by considering the selected options $v_1, \ldots, v_i$ within different sub-families.

**Incomplete refinement strategy** We suggest the following incomplete refinement strategy to focus the search (at the cost of completeness). In particular, we restrict the exploration to FSCs that are structurally close to $\pi^*$ as follows: we i) fix the options selected by $\pi^*$ in perfectly observable states, ii) fix the options in the consistent parameters, and iii) remove options in the inconsistent parameters that were not selected by $\pi^*$, i.e., the set $V_{(n,z)} \setminus \{v_1, \ldots, v_i\}$.

### 3.3 COUNTEREXAMPLE-BASED TEACHER

The MDP abstraction employs deductive reasoning: It talks about a set of FSCs at once to deduce conclusions about the individual members of this set. In this subsection, we discuss the orthogonal, inductive, approach. We suggest a (smart) enumeration over individual FSCs inspired by Ceska et al. [2021]. If the FSC is satisfactory, i.e., it is admissible and has good value, this helps the teacher. Otherwise, if the FSC is not satisfactory, we learn facts, called *counterexamples*, that help us to avoid considering other FSCs.

To realise this teacher, we represent the FSCs that have not been pruned as a propositional formula[6]. We use the SMT solver CVC5 [Barbosa et al., 2022] (over quantifier-free bounded integers) to effectively manipulate the propositional formula and to find FSCs that have not been pruned.

We assume a constraint $P_{\geq \lambda}(\Diamond T)$, a family $\mathcal{F}$, the state-

---

vector $ub$ obtained from the maximising policy $\pi^*$ in MDP $\mathcal{A}^{\mathcal{F}}$ as discussed in Sec. 3.2, and an FSC $F \in \mathcal{F}$.

[6] For each parameter $k$, there is a corresponding integer variable $x_k$ whose domain $V_k$ corresponds to all possible realisations of $k$. Initially, the design space is encoded as a conjunction $\wedge_{k \in K} : x_k \in V_k$. Pruning a subfamily corresponds to adding to this formula a negated clause describing all the pruned realisations. Details can be found in [Ceska et al., 2021].

**Definition 4.** A *counterexample* (CE) for FSC $F$ and $P_{\geq \lambda}$ is a subset $C \subseteq S^F$ that induces the sub-MC of $\mathcal{M}^F$ given as $\mathcal{M}_C^F = (C \cup \text{succ}(C) \cup \{s_\perp, s_\top\}, (s_0, n_0), P')$ with $P'(s) =$

$$
\begin{cases}
P^F(s) & \text{if } s \in C, \\
[s_\top \mapsto ub(s), s_\perp \mapsto 1-ub(s)] & \text{if } s \in \text{succ}(C) \setminus C, \\
[s \mapsto 1] & \text{if } s \in \{s_\top, s_\perp\},
\end{cases}
$$

where $\text{succ}(C)$ is the set of direct successors of $C$, and the probability to reach $T \cup \{s_\top\}$ in $\mathcal{M}_C^F$ is $< \lambda$.

Intuitively, in the sub-MC, states $s$ outside the CE $C$ evolve to $s_\top$ with probability $ub(s)$, the maximal probability to reach $T$ from $s$ in the family $\mathcal{F}$ (i.e., the worst-case possible in $\mathcal{F}$). They evolve to $s_\perp$ with probability $1-ub(s)$, the minimal probability to avoid $T$ in $\mathcal{F}$. For $(s, n) \in C$, the parameter $(n, O(s)) \in K$ is called *relevant*. The CE for the constraint $P_{\leq \lambda}$ is defined similarly using $lb$ rather than $ub$.

For each $F' \in \mathcal{F}$ that for each relevant parameter in a CE $C$ uses the same values as $F$, it holds that $P^{F'} < \lambda$. Therefore, we can safely remove $F'$ from the design space. We say that $C$ *generalised* to the set of all such $F'$.

Smaller CEs lead to generalisation to larger families of FSCs. As computing minimal CEs is NP-complete [Funke et al., 2020], we adopt the greedy approach from [Andriushchenko et al., 2021a]. Handling multiple constraints is straightforward as we can compute the CE for each constraint violated by the FSC $F$. This can potentially improve the pruning.

Similarly as the incomplete refinement strategy in Sec. 3.2, we consider an incomplete generalisation of the CEs. In particular, we redefine the notation of relevant parameters. The parameter $(n, O(s))$ for $(s, n) \in C$ is relevant only if the observation $O(s)$ is inconsistent in $\mathcal{A}^{\mathcal{F}}$ or the option selected by $F$ is different from the options selected by $\pi^*$. This leads to more aggressive pruning and restricts the exploration to the FSCs that are topologically close to $\pi^*$.

**Example 4.** Consider a variant of our maze problem with initial state $s_0$, family $\mathcal{F}$ of all 1-FSCs, where the available set of actions in the observation $o_3$ is restricted s.t. $V_{(n_0, z_3)} \in \{u, d, r\} \times \{0\}$. Let FSC $F$ as in Fig 3 (left). The right part illustrates the induced MC and the middle part shows the CE $C$ for the constraint $P_{\geq 1}$. Note $P((s_4, n_0), s_\perp) = 1$ as $ub(s_4) = 0$. Thus, the relevant parameters are $(n_0, z_i)$ for $i \in \{0, 1, 2\}$. The generalisation of $C$ enables pruning a significant part of $\mathcal{F}$. Under incomplete generalisation, the parameter $(n_0, z_0)$ is not relevant as it is consistent in $\mathcal{A}^{\mathcal{F}}$ and $F$ picks the same option as $\pi^*$.

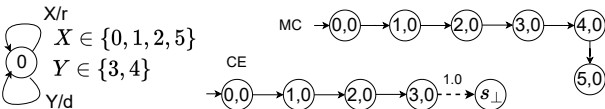

Figure 3: A CE for the given FSC in the maze problem.

# 4 MEMORY INJECTION STRATEGY

This section discusses the outer stage of our approach, cf. Fig. 1, in which the learner decides *where* to search. In particular, a subset of FSCs is selected and passed onto the teacher, as outlined in Sec. 3. We assume access to an abstraction oracle that yields bounds on the value for every state based on an abstraction scheme outlined in Sec. 3.2. The learner processes this information and derives a new design space. It does so by combining three ingredients: 1. *Adding memory*: By allowing FSCs to store more information, we (drastically) increase the design space. We allow to locally increase memory to keep the growth manageable. 2. *Removing symmetries*: Similar to [Grześ et al., 2013], it is unnecessary to include symmetric FSCs in the design space. 3. *Analysing abstractions*: Use the results from the abstraction to guide the search.

## 4.1 ADDING MEMORY

The key idea of the memory injection strategy is to use the diagnostic information obtained from the preceding inner loop exploring the design space represented by a family $\mathcal{F}$ to construct a new family $\mathcal{F}'$, say. These families are based on two fixed memory structures (either as a full or reduced FSC). On constructing $\mathcal{F}'$, memory can be added that corresponds to one of the observations, see Sec. 3.1. This section outlines where to add the memory.

To decide how to extend the family $\mathcal{F}$, we use the following information: 1. The maximising policy $\pi^*$ in MDP $\mathcal{A}^{\mathcal{F}}$ together with its corresponding bounds $ub$[7]. 2. The FSC $F^*$ in $\mathcal{F}$ obtained from the teacher. That FSC is not available if the FSC is inadmissible or if the teacher is aborted.

If only $\pi^*$ is available, the memory injection strategy employs a similar idea as the refinement strategy described in Sec. 3.3. In particular, it evaluates the significance of the inconsistent observations w.r.t. $\pi^*$ by aggregating the significance of their inconsistent parameters as described above. By adding the memory to the most significant inconsistent observation, we try to resolve the inconsistency and drive the search towards an FSC that mimics $\pi^*$ as often as possible, i.e., along most paths.

[7]The state space of MDP $\mathcal{A}^{\mathcal{F}}$ includes copies of states where memory has been added. Without symmetry reduction, an optimal policy (mostly) takes the same action in these copies, thus making the copies redundant. However, in combination with the symmetry reduction, the outgoing transitions of these copies differ and the copies are then no longer redundant.

If additionally $F^*$ is available, the idea is similar: to obtain a better FSC in the next iteration, we identify the observation in which the choices of $F^*$ differ from the observation-free policy $\pi^*$ the most. That is, the inconsistency measure $\chi(s)$ for state $s$ is now the absolute difference in $ub(s)$ (for the maximising property) w.r.t. two actions: the one provided by $\pi^*$ and the one from $F^*$. The inconsistency measure of the observation $z$ is now the weighted average of $\chi(s)$ across all states $s$ with $O(s) = z$, where the weight for state $s$ is the expected number of visits of $s$ in the FSC $F^*$.

Note that in both cases the proposed inconsistency measure is just a heuristic and neither guarantees that adding memory to the selected observation will improve the value of the FSC, nor that the memory injection strategy will eventually find the optimal $F^*$. In fact, without additional modifications to the strategy, the heuristic will keep adding memory to a single observation because adding memory does not *resolve* the inconsistencies. To mitigate this problem, it is crucial to employ the symmetry reduction.

## 4.2 SYMMETRY REDUCTION

Adding memory to a selected observation typically creates a family $\mathcal{F}$ that includes FSCs having the same value due to certain symmetries in their topology. Removing these symmetries from $\mathcal{F}$ reduces: i) the size of $\mathcal{F}$ and ii) the number of inconsistencies of the policy $\pi^*$ obtained from $\mathcal{A}^{\mathcal{F}}$, see below. As a result of resolving these inconsistencies, the memory injection strategy is capable of better recognising where *else* in the POMDP memory is needed. The importance of symmetry breaking has been recognised by Grześ et al. [2013], who proposed a generation strategy for isomorphism-free Moore automata. We propose a different approach based on restricting the family of candidate FSCs. We illustrate where the symmetries are introduced as well as how to deal with them in the example below.

**Example 5.** Consider again the maze problem from Example 1 and the specification to minimise the expected number of steps to reach $T$. Note that this includes an implicit constraint to reach $T$. Let $\mathcal{F}_1$ be the family of all 1-FSCs (no memory was added). The inner loop detects that there is no admissible 1-FSC satisfying the constraint: in observation $z_1$ we need to be able to pick both actions $r$ and $l$, and similarly for observation $z_4$. Assume that the minimising policy $\pi_1^*$ in $\mathcal{A}^{\mathcal{F}_1}$ reveals that the most significant inconsistency is the one in observation $z_1$. Adding memory to $z_1$ has a twofold effect on the resulting design space $\mathcal{F}_2$.

First, a new parameter $(n_1, z_1)$ is introduced that encodes action selection in newly created copies of states with observation $z_1$. Second, each state having successor $s$ with observation $z_1$ must be able to choose whether to go to $(s, n_0)$ or its copy $(s, n_1)$. In our case, the domains of parameters $(n_0, z_0)$, $(n_0, z_2)$, $(n_0, z_3)$ as well as of parameters $(n_0, z_1)$ and $(n_1, z_1)$ (remember the self-loops) will now be

$\{u, r, d, l\} \times \{n_0, n_1\}$.

Before proceeding with the inner loop, we first remove symmetric assignments from the family $\mathcal{F}_2$. That is, consider FSC $(\gamma, \delta)$ with $\gamma(n_0, z_1) = l$ and $\gamma(n_1, z_1) = r$. Clearly, this FSC achieves the same value as the symmetric FSC $(\gamma', \delta')$ with $\gamma'(n_0, z_1) = r$, $\gamma'(n_1, z_1) = l$ and $\delta'(\cdot, z) = n_1$ if $\delta(\cdot, z) = n_0$ and vice versa, for each predecessor observation $z$ of $z_1$. The family $\mathcal{F}_2'$ should include only one of the two FSCs, so we modify the domains of parameters $(\cdot, z_1)$ as follows: $V_{(n_0, z_1)} = \{u, d, r\} \times \{n_0, n_1\}$ and $V_{(n_1, z_1)} = \{u, d, l\} \times \{n_0, n_1\}$.

The inner loop again detects that no admissible solution exists. Minimising policy $\pi_2^*$ now contains a single inconsistency in observation $z_4$, which we amend by adding memory to $z_4$. In the resulting family $\mathcal{F}_3$, we introduce parameter $(n_1, z_4)$ for action/memory selection in the new copy, and modify parameters $(n_0, z_5), (n_0, z_4)$ and $(n_1, z_4)$ to enable the transition to the newly created copies (other successors of observation $z_4$ already account for both possible memory updates). As in the previous case, we break the symmetry in action selection in $(\cdot, z_4)$: $V_{(n_0, z_4)} = \{u, r, l\} \times \{n_0, n_1\}$ and $V_{(n_1, z_1)} = \{d, r, l\} \times \{n_0, n_1\}$. The third iteration of the inner loop finally yields an optimal FSC with value $7.1\overline{6}$. No additional memory injection can improve upon this.

More generally, for each observation $z$, we keep a list $\mathcal{I}_{Act}^z$ of actions in which parameters $(\cdot, z)$ were inconsistent. This list is updated each time we update memory for observation $z$. Upon adding memory to $z$, we apply the symmetry reduction to the corresponding domains of parameters $(n_i, z)$. For simplicity, let $|\mathcal{I}_{Act}^z| = \mu(z)$, as in Example 5. In such a case, $V_{(n_i, z)}$ is set to $(Act(z) \setminus \mathcal{I}_{Act}^z \cup \{\mathcal{I}_{Act}^z[i]\}) \times N$, where $\mathcal{I}_{Act}^z[i]$ is the $i$-th action in the list $\mathcal{I}_{Act}^z$. This ensures that for each action $a \in \mathcal{I}_{Act}^z$ there is exactly one parameter $V_{(\cdot, z)}$ where $a$ is available. If $|\mathcal{I}_{Act}^z| \neq \mu(z)$, the construction of domains $V_{(\cdot, z)}$ becomes more involved: one must take possible inconsistencies in memory updates into account. For a comprehensive description of the symmetry reduction, we refer to the implementation (see Sec. 5). In general, there are two extreme ways one can proceed with the symmetry reduction. Either one can enable all of the actions/memory updates in $V_{(\cdot, z)}$, which will undermine the observation selection discussed in Sec. 4.1. The other extreme is to disable choices in $V_{(\cdot, z)}$ arbitrarily. This may lead to incompleteness, as the following example shows.

**Example 6.** Assume the following modification of Example 1, where an agent now experiences a slight drift to the west: upon choosing the direction, the agent will move to the selected direction with probability 0.9 and will otherwise move one cell to the left (if available). For instance, when moving down from the state 2, the agent might instead end up in the state 1. FSC synthesis proceeds similarly as in Example 5, so assume that after two memory injections and symmetry reductions we end up with the same family $\mathcal{F}_2'$ having $V_{(n_0, z_1)} = \{u, d, r\} \times$

$\{n_0, n_1\}$, $V_{(n_1, z_1)} = \{u, d, l\} \times \{n_0, n_1\}$ and $V_{(n_0, z_4)} = \{u, r, l\} \times \{0, 1\}$, $V_{(n_1, z_1)} = \{d, r, l\} \times \{0, 1\}$. The optimal assignment for the state 2 is $\gamma(n_0, z_2) = d$ (moving down) and $\delta(n_0, z_2) = n_1$, since $(n_1, z_4)$ is the only parameter that enables movement down from the state 6. However, if due to drift the agent ends up in state $(s_1, n_1)$, it cannot move right $(r \notin V_{(n_1, z_1)})$ and is forced to move sub-optimally. In our case, we could have preserved the optimal solution if we had switched the order of e.g. second symmetry reduction, although predicting a non-conflicting reduction order would require additional analysis.

## 5 EXPERIMENTAL EVALUATION

Our evaluation focuses on the following questions:

*Q1: How does our approach compare with state-of-the-art belief-based approaches?* Belief-based approaches are the widespread approach to solving POMDPs. They implicitly or explicitly approximate the large or infinite belief-MDP instead of searching for policies with a particular structure. We compare with the approaches by Norman et al. [2017] (implemented in PRISM Kwiatkowska et al. [2011]) and by Bork et al. [2022] (implemented in Storm [Dehnert et al., 2017]). These methods provide state-of-the-art techniques for finding policies in belief MDPs for indefinite-horizon specifications, i.e., without discounting.

*Q2: How does our approach compare to state-of-the-art approaches to synthesise deterministic FSCs?* To this end, we qualitatively compare with the state-of-the-art dual MILP formulation from [Kumar and Zilberstein, 2015] which uses a max-entropy strategy for adding memory nodes. We also consider a recent alternative formulation of a primal MILP in [Winterer et al., 2020] for multi-objective specifications.

*Q3: What is the effect of our heuristics on the run-time and the value of the resulting FSCs?* We discuss additional insights from an ablation study in which we discuss which settings yield the best performance.

**Selected benchmarks and setup** The framework outlined above has been implemented in PAYNT Andriushchenko et al. [2021b], a tool for inductive synthesis of probabilistic programs[8]. Unless mentioned otherwise, we used benchmarks from [Bork et al., 2020, 2022] extended by a few more involved variants. Table 1 lists the statistics of the models including the number of states, the overall number of actions, and the number of observations. Our experiments run on a single core of a machine equipped with an Intel i5-12600KF @4.9GHz CPU and 32 GB of RAM. An artefact allowing one to reproduce our experimental evaluation is available at `https://doi.org/10.5281/zenodo.6637489`.

**Threads to validity** This evaluation focuses on showing the *potential* of our approach using benchmarks from

---

[8]See `https://github.com/randriu/synthesis`

the verification literature. While the comparison with implementations in Storm and PRISM is thorough, other algorithms were not available and thus we resort to comparing with the performance reported in those papers. There is a dire need for a more structural comparison over different algorithms. Furthermore, the experiments suffice to provide insights in the performance of the algorithm, but not to draw conclusions about the relative relevance of individual heuristics. Finally, while our approach is general and could be applied to queries such as expected discounted rewards, it can only be competitive if it is tailored to that setting.

**Q1: Comparison to belief-based methods**

Table 1 summarises key experimental results related to **Q1**. The columns list the following information (from left to right): the model and its variant, the model statistics, the bounds provided by [Norman et al., 2017] and its run-time, the lower bounds provided by [Bork et al., 2022] and its run-time (for two settings: the fastest synthesis and the best bound), the results provided by our approach (including the number of added memory nodes) and its run-time (the first interesting solution and the best solution found), and the upper bounds provided by Bork et al. [2020] allowing us to judge the quality of the lower bounds.

To simplify the presentation, this table shows results achieved by our approach using the default setting (different settings are used for the entries denoted by *): the inner loop is instantiated by the pure MDP abstraction oracle with the incomplete refinement strategy, and the outer loop uses the memory injection strategy with symmetry reduction. The impact of our optimisation heuristics as well as the results for multi-objective specifications are discussed under **Q3**.

The results demonstrate that *our inductive approach is competitive with the belief-state space approximation for indefinite-horizon specifications*. For models with a moderate number of observation/actions, we provide better trade-offs between the run-time and the values of the found policies. Moreover, we found small FSCs that improve the lower-bounds in [Bork et al., 2022]. For models with a large number of observations/actions, we found small high-value FSCs in comparable run-time. For the *Rocks* model and a larger *Netw* model, we failed to find a good solution. We highlight two interesting results: For *Grid-av 4-0*, our strategy injected four memory nodes (see †) and achieved the bound provided by the observation-free MDP abstraction which guarantees the global optimum. For *Drone 4-2*, we found a very small FSC that achieves the known upper bound on the solution value [Bork et al., 2020].

**Q2: Comparison to MILP-based FSC synthesis**

A direct comparison with MILP-based approaches is complicated due to limited availability of standardised imple-

mentations. Qualitative comparisons are furthermore complicated by differences in benchmarks.

However, based on the *Hallway* model from [Kumar and Zilberstein, 2015] and manually translating it to mimic the effect of discounting and stochastic rewards to an almost equivalent model [9], we make the following preliminary observation. The dual MILP optimisation for the fixed-size reactive FSC (equivalent to our 1-FSC) achieved the value 0.32 in 15 minutes. Given the discount factor 0.95, this corresponds to an FSC where the expected number of steps to reach the target equal to $\log_{0.95}(0.32) = 22.2$. Using the memory injection strategy, they found an FSC with 14 additional memory nodes in $\sim$1 hour achieving value 0.46, i.e., 15.1 expected steps. Our complete strategy explored all 1-FSCs in less than 1s and found a solution achieving 18.5 expected steps. The restricted exploration of full 4-FSCs found a solution achieving 14.9 expected steps in 218s. The default strategy used in Table 1 (see above) added one memory node and found a solution achieving 16.1 expected steps in less than 1s. Despite all the limitations of this experimental setup, *these results indicate that our approach can be at least as successful as MILP-based synthesis methods.*

Similarly, we compare with [Winterer et al., 2020] on the *Grid-av 4-0* model with a constraint on the reachability probability and the minimisation of an reward. The best solution of the MILP optimisation with a restricted randomisation and memory injection has value 3.43 (found within seconds). This solution is obtained by our default strategy within 1s by adding one memory node. In 21s, it added seven memory nodes and found a better solution having value 3.29. This shows that *our inductive approach outperforms the MILP optimisation also on multi-objective specifications.*

**Q3: The effect of optimisation heuristics**

*Efficacious heuristics:* We generally remark that the design spaces in this paper are several orders of magnitude bigger than the design spaces supported by the more general-purpose inductive synthesis framework in [Andriushchenko et al., 2021b]. The superior performance can mostly be explained by the tailored representation of the design space and novel search heuristics.

*Incomplete search:* To find a good policy, it is not necessary to be complete. The (default) incomplete refinement strategy and CE generalisation is beneficial for handling large number of observations/actions. The complete exploration fails to find a good solution for, e.g., the *Drone* models. In our experiments, we did not observe that the incomplete exploration discards important solutions except the *Grid-av 4-0.1* model. For that model, the incomplete strategy performs 10 memory injections and finds in 840s a solution with value 0.92. The complete strategy performs 8 memory injections

---

[9]The values of the resulting FSCs are comparable.

| Benchmark | | Size | | PRISM | Storm | | Inductive synthesis | | Upper- |
| Model | Spec. | $S$/$Act$ | $Z$ | | first | best | fastest | best | bounds |
|---|---|---|---|---|---|---|---|---|---|
| Grid-av 4-0 | $P_{\max}$ | 17 59 | 4 | [0.21,1] <1s | 0.86 <1s | **0.93** **<1s** | **0.93 (3)** **<1s** | 0.93(4)$^\dagger$ <1s | ≤ 0.98 |
| Grid-av 4-0.1 | $P_{\max}$ | 17 59 | 4 | [0.21,1] <1s | 0.82 <1s | 0.85 124s | 0.92 (4) <1s | **0.93 (5f)** **53s*** | ≤ 0.99 |
| Grid 30-sl | $R_{\min}$ | 900 3587 | 37 | TO/MO | 121 1s | - - | **119 (6)** **150s*** | - - | ≥ 116.1 |
| Maze sl | $R_{\min}$ | 15 54 | 8 | [**7.09**,7.09] **2s** | 7.67 <1s | - - | **7.14 (3)** **<1s** | **7.09 (3f)** **1s*** | ≥ 7.08 |
| Crypt 4 | $P_{\max}$ | 1972 4612 | 510 | [0.33,0.79] 6s | **0.33** **<1s** | - - | **0.33 (0)** **<1s** | - - | ≤ 0.33 |
| Nrp 8 | $P_{\max}$ | 125 161 | 41 | [0.13,0.24] 3s | **0.13** **<1s** | - - | **0.13 (0)** **<1s** | - - | ≤ 0.13 |
| Hallway | $R_{\min}$ | 61 301 | 23 | TO/MO | 19.3 <1s | 19.2 <1s | **16.3 (1)** **<1s** | **14.9 (4f)** **218s*** | ≥ 12.4 |
| Drone 4-1 | $P_{\max}$ | 1226 3026 | 384 | TO/MO | 0.79 <1s | - - | 0.71 (0) 1s | **0.87 (2)** **915s** | ≤ 0.94 |
| Drone 4-2 | $P_{\max}$ | 1226 3026 | 761 | TO/MO | 0.86 <1s | 0.91 138s | 0.94 (0) <1s | **0.97 (2)** **326s** | ≤ 0.97 |
| Refuel 6 | $P_{\max}$ | 208 565 | 50 | [0.67,0.72] 136s | **0.67** **<1s** | - - | 0.44 (2) <1s | 0.67 (2f) 45s* | ≤ 0.69 |
| Netw-p 2-8-20 | $R_{\max}$ | $2\cdot10^4$ $3\cdot10^4$ | 4909 | [**557**,557] **1099s** | 537 <1s | - - | 540 (0) 105s | - - | ≤ 558 |
| Rocks 12 | $R_{\min}$ | 6553 $3\cdot10^4$ | 1645 | TO/MO | 38 <1s | **20** **47s** | 42 (0) 1s | - - | ≥ 20 |

Table 1: Results for **Q1**. Bold entries denote the best solutions, – indicates that no better solution was found within 30 minutes, * indicates that non-default settings were used, TO/MO denotes timeout (30 minutes) or out of memory.

and finds in 1189s a solution with value 0.93. For this benchmark, the best synthesis result reported in Table 1 (column *best*) relied on complete exploration of full 5-FSCs.

*Memory injection:* this prevents the blowup that just increasing memory nodes has. Without symmetry reduction, the abstraction-based framework has trouble guiding the search. Symmetry reduction thus not only reduces the design space but it also guides the memory injection strategy correctly select the most promising observation. For example, in the *Maze sl* model, the memory injection without symmetry reduction repeatedly adds memory to a single observation and the optimal solution is not found. On the other hand, symmetry reduction can discard an optimal solution as demonstrated on the *Grid-av 4-0.1, Maze sl, Hallway* and *Refuel 6* models. For these models, the column *best* of Table 1 lists the results of the exploration of full $k$-FSCs denoted as $k$f for $k \in \{2, 3, 4, 5\}$.

*Hybrid teacher:* For the models in Table 1, the use of CEs or a hybrid teacher (combining MDP abstraction and CE pruning) does not improve the synthesis process. We believe that future work towards CE-guided inductive synthesis may change this balance. As of now, only for models where the MDP abstraction is significantly larger than the induced MCs corresponding to the candidate FSCs, the hybrid approach is superior: e.g., for the *Grid 30-sl* model, a larger variant of the grid-like model, the MDP abstraction is 15x larger than the individual induced MCs. With default set-

tings no admissible FSC is found within 30 minutes. A hybrid teacher helps finding an FSC within 150s that improves the solution found by the belief-based method.

*Multi-objective (MO) specifications:* Apart from the MO variant of the *Grid-av 4-0* model discussed in **Q2**, we also considered a MO variant of the *Maze sl* model including a more complicated specification with an additional reach-avoid constraint. The constraint restricts the optimal FSC, but the run-time of the synthesis remains < 1s.

# 6 CONCLUSION

This paper presents a first inductive-synthesis based framework for finding finite-state controllers (FSCs) in POMDPs. Key ingredients are the novel heuristics to incrementally construct the memory structure of the FSC as well as two oracles for searching and evaluating families of FSCs. The experimental results show promising results indicating that this framework is competitive with the state-of-the-art alternatives. Future work includes the integration of belief-based approaches as an additional oracle.

**Acknowledgements**

This work has been supported by the Czech Science Foundation grant GJ20-02328Y and the ERC AdG Grant 787914 (FRAPPANT). The authors thank Alexander Bork and Filip Macák for their support in running the experiments.

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
