# OpenReview forum: "Inductive Synthesis of Finite-State Controllers for POMDPs"
_auai.org/UAI/2022/Conference — UAI 2022 Poster_

### Official Review · Reviewer_EQeE · 2022-03-24

**Q2(1) Originality/Novelty:** 3
**Q2(2) Significance/Impact:** 3
**Q2(3) Correctness/Technical Quality:** 3
**Q2(6) Clarity Of Writing:** 3
**Q6 Overall Score:** 7
**Q8 Confidence In Your Score:** 1

**Q1 Summary And Contributions:**

The paper presents a two-layered approach to synthesizing finite-state controllers (FSC) for partially observable Markov decision processes (POMDP) . The  out layer, called the learner, is responsible for the construction of a design space (i.e., a finite family of FSCs with a fixed number of nodes) and the inner layer, called the teacher, will evaluate FSCs in the space and find the best one within it.  The approach is implemented and compared with existing methods in experimental evaluations.


**Q2 Assessment Of The Paper:**

More detailed information regarding each of these aspects is given below:

**Q2(4) Quality Of Experiments (Optional):**

3: Good: The experimental evaluation is adequate, and the results convincingly support the main claims.

**Q2(5) Reproducibility:**

2: Fair: Key resources (e.g., proofs, code, data) are unavailable but key details (e.g., proof sketches, experimental setup) are sufficiently well-described for an expert to confidently reproduce the main results.

**Q3 Main Strengths:**

The proposed approach is novel. The paper is well-organized and presented. Although it is somewhat difficult for non-experts to follow the details of the method, the explanation of general ideas behind the system is conceptually easy to understand. In addition, experimental comparisons with state-of-the-art  approaches on a variety of benchmarks demonstrate the superiority of the proposed approach.

**Q4 Main Weakness:**

I can not find serious weakness of the work. Perhaps, a missing point is about the calculation of optimal policy. It seems that the optimal policy \pi^* plays a crucial role in the inner stage and the refinement strategy (Sec3.2), but nothing is said about how to find it. In addition, there are some undefined notions or typos to be clarified (see Q5).

**Q5 Detailed Comments To The Authors:**

The work may be valuable itself, but the presentation could be further improved by clarifying several points.

1. In the definition of FSC, it is said that nodes represent states. Then, in the description of the algorithm, they are called memory nodes and the number of nodes are identified with the amount of memory used in each observation. I cannot see the connection. It may be helpful to elaborate on this point.
2. As mentioned above in Q4, it is unclear how an optimal policy for the abstraction of MCs can be found. Is it an easy task or does it have any impact on the complexity of the algorithm?
3. After Definition 1, it is said that the family of full FSCs contain O(|N||Z|) many FSCs. However, |N||Z| is the size of the domain (or the number of parameters in the terminology of the paper). Considering different ways of assigning values to these parameters, there should be \Pi_{(n,z)\in K}|V_{(n,z)}| different functions, which seemingly correspond to the different FSCs in the family.
4. Analogously, I cannot see why the number of possible mappings in the family of reduced FSCs is O(|N|+|Z|). It seems that the size of K is |K|=\sum_{z\in Z}\mu(z) according to Definition 2. Hence, the number of parameters may decrease in this case, but the formula for the number of different FSCs should remain unchanged.
4. In Sec.3.3, it is said that un-pruned FSCs are represented as a propositional formula. How?
6. Typos:
- L. 1&2 of Sec.2: X or S?
- P.3, L.1 for the definition of POMDP: Z(s)=z should be O(s)=z
- Footnote 4: thxe -> the
- Example 2: j\in\{u,d,l,r\} -> j\in\{0,...,5\}
- P.4, L.3, right column: the codomain of \pi is Act or Act^{\cal F}?
- Definition 4: The exact definition of direct successor of a state (or a set of states) is not given. Does it mean that s' is a successor of s if P(s'|s)>0?

**Q7 Justification For Your Score:**

The novelty of the proposed approach and its performance on the experimental evaluation appears sufficient for justifying the acceptance of the paper.

**Q9 Complying With Reviewing Instructions:**

1: Yes.

---

### Official Review · Reviewer_DPb2 · 2022-04-11

**Q2(1) Originality/Novelty:** 2
**Q2(2) Significance/Impact:** 2
**Q2(3) Correctness/Technical Quality:** 3
**Q2(6) Clarity Of Writing:** 3
**Q6 Overall Score:** 5
**Q8 Confidence In Your Score:** 2

**Q1 Summary And Contributions:**

The contribution considers finite, partially observed, MDPs, and puts forward an inductive synthesis approach to generate deterministic, finite-state controllers that can attain given specifications.



**Q2 Assessment Of The Paper:**

More detailed information regarding each of these aspects is given below:

**Q2(4) Quality Of Experiments (Optional):**

4: Excellent: The experimental evaluation is comprehensive and the results are compelling.

**Q2(5) Reproducibility:**

3: Good: Key resources (e.g., proofs, code, data) are available and key details (e.g., proofs, experimental setup) are sufficiently well-described for competent researchers to confidently reproduce the main results.

**Q3 Main Strengths:**

-- To the best of my knowledge, some novelty, at least in combining ingredients
-- Experimental evaluation looks thorough (again, having not checked carefully)
-- Good explanation of the underlying theory
-- I find the paper readable

**Q4 Main Weakness:**

-- Appears (again, as an outsider) incremental
-- Seems like a bulls eye paper for a mainstream verification conference or a venue like QEST more focused on quantitative aspects of verification. If reviewed there,  the comparison with prior work would be reviewed by hardcore probabilistic verification experts. Not sure if that will happen at UAI.

**Q5 Detailed Comments To The Authors:**

It would be good to clarify how much impact partial observability has on the set up (yes, it does play a role, but is it a big one?)

The contribution considers finite, partially observed, MDPs, and puts forward an inductive synthesis approach to generate deterministic, finite-state controllers that can attain given specifications.

The crux of the approach lies on the inductive engine for synthesis, which consists of two interleaved loops, an outer one that generates and updates a design space, and an inner one that attempts FSC synthesis within this space. The engine leverages a number of oracles enabling these two searches.
A number of tools are used to reduce the search space -- e.g. a state space abstraction that allows one to rule out certain areas of the space.
these tools are all reminiscent of standard techniques in probabilistic verification. But the paper appears to be a step forward, and the experimental results compare with state of the art tools like PRISM.

In terms of evaluation, a disclaimer is that I am far from an expert in the area.  And I know that comments of the form "this paper should really have been sent to Y instead of X"  are often a cop out for a referee who has nothing better to say. Which honestly does apply here.  That being said, here goes: I wonder why the paper is not sent to a verification conference. It is a contribution that is some increment (moderate to small) on top of the state of the art finite state synthesis techniques. And many of the experts -- certainly the main competitors  (e.g. PRISM/STORM) considered here -- are mainstays of the verification community. I am sure UAI accepts this kind of paper; it can be easily cast as AI. Although these days everything is AI. If there is someone who is more conversant with the increment over the prior art, I would of course defer to them.

**Q7 Justification For Your Score:**

Questions of depth of contribution and comparison to related work (see above)

**Q9 Complying With Reviewing Instructions:**

1: Yes.

---

### Official Review · Reviewer_RX17 · 2022-04-12

**Q2(1) Originality/Novelty:** 3
**Q2(2) Significance/Impact:** 3
**Q2(3) Correctness/Technical Quality:** 3
**Q2(6) Clarity Of Writing:** 4
**Q6 Overall Score:** 7
**Q8 Confidence In Your Score:** 3

**Q1 Summary And Contributions:**

In their paper, the authors make a suggestion for obtaining a finite-state controller from a POMDP. The underlying idea is to make use of refinements, counter-example guided pruning and several strategies like adding memory and removing symmetries for this purpose.

**Q2 Assessment Of The Paper:**

More detailed information regarding each of these aspects is given below:

**Q2(4) Quality Of Experiments (Optional):**

3: Good: The experimental evaluation is adequate, and the results convincingly support the main claims.

**Q2(5) Reproducibility:**

3: Good: Key resources (e.g., proofs, code, data) are available and key details (e.g., proofs, experimental setup) are sufficiently well-described for competent researchers to confidently reproduce the main results.

**Q3 Main Strengths:**

+ Innovative approach for controller synthesis for POMDPs
+ Well written and structured paper
+ The experimental analysis is appropriate and shows improvements


**Q4 Main Weakness:**

- The threats to validity are not discussed in detail

**Q5 Detailed Comments To The Authors:**


The paper is well written and structured. The authors guide the reader through the paper and there are always examples given. Although, the used principles for improving synthesis might be well-known to a certain extent. Their use in the particular context seems to be innovative and of interest. Besides the foundations, the authors report on results obtained from the experimental analysis. The analysis is based on a larger set of models, and is well described and carried out. The obtained results show that the proposed approach improves controller synthesis compared with other baseline approaches. There is only one drawback: The comparison is based on different implementations. Hence, runtime comparisons might depend very much on the implementations, which can hardly be avoided. However, it would be sufficient to mention threats to validity in the paper.


**Q7 Justification For Your Score:**

The paper is well written and comprises all parts that are necessary to understand the underlying concepts and the proposed approach. The experimental evaluation is fine comprising enough models and a comparison with previous baseline approaches.

**Q9 Complying With Reviewing Instructions:**

1: Yes.

---

### Official Review · Reviewer_2uBd · 2022-04-13

**Q2(1) Originality/Novelty:** 2
**Q2(2) Significance/Impact:** 2
**Q2(3) Correctness/Technical Quality:** 3
**Q2(6) Clarity Of Writing:** 2
**Q6 Overall Score:** 4
**Q8 Confidence In Your Score:** 4

**Q1 Summary And Contributions:**

The paper proposes a new method for finding policies for POMDPs in the form of finite-state controllers. The method is based on iteratively finding better and better FSCs, alternating between setting up a search space based on last found candidate FSC, and finding a new better FSC.


**Q2 Assessment Of The Paper:**

More detailed information regarding each of these aspects is given below:

**Q2(4) Quality Of Experiments (Optional):**

2: Fair: The experimental evaluation is weak: important baselines are missing, or the results do not adequately support the main claims.

**Q2(5) Reproducibility:**

2: Fair: Key resources (e.g., proofs, code, data) are unavailable but key details (e.g., proof sketches, experimental setup) are sufficiently well-described for an expert to confidently reproduce the main results.

**Q3 Main Strengths:**

Experimentally the proposed approach seems to be quite strong in comparison to related earlier work.



**Q4 Main Weakness:**

Large parts of the paper are not properly readable, as only a quite abstract but complicated explanation is given, with limited technical details. E.g Section 4 does not contain sufficient detail to allow reconstructing what is being done.

The experimental evaluation is narrow.

**Q5 Detailed Comments To The Authors:**


No clear reason is given why limitation to indefinite-horizon problems has been adopted. The ideas would be equally applicable to other classes of problems, for which very strong competing approaches exist.

*Q5 Detailed Comments To The Authors
Please provide constructive criticism and feedback that could help improve the work or its presentation (e.g., presentation suggestions, missing references, minor mistakes and typos or grammar improvements). You may also include questions to the author here.

The inductive synthesis framework has similarities with various counter-example guided abstraction refinement (CEGAR) approaches used in verification and validation: your limited FSC "design space" is the "abstraction", and the design space is refined (made larger / less abstract) when a not-completely-satisfactory solution candidate emerges, and the two stages, finding a solution under an abstraction, and refining the abstraction, are alternated. You could point out this similarity.

On page 5 typo "quantified-free"

End of page 1: in "adapts the design space" the word "adapts" does not have the right connotations. What is the design space "adapted" to? "Adaptation" presupposes some cause for the change. Maybe the word "modify" were better?

I don't like the terminology "learner" and "teacher" and "oracle". Is this really somehow "multi-agent" approach, or are you just unnecessarily antropomorphizing parts of your algorithm?

"Finding optimal deterministic FSCs is ETR-complete": Finding something is not a decision problem, and hence the NP-completeness should be explained better here. Or, alternatively, drop the mentions of the complexities.
Same issue with "NP-complete" on page 5: only decision problems can be NP-complete. You can say that a function problem is NP-hard, but then you have to state the "in NP" part differently, as NP consists of decision problems only.


**Q7 Justification For Your Score:**

Comparison to related work is narrow.

Experimental comparison is too narrow.


**Q9 Complying With Reviewing Instructions:**

1: Yes.

---

### Decision · Program_Chairs · 2022-05-15

**Decision:**

Accept (Poster)

**Comment:**

Meta Review: The paper presents a novel learning framework (with a concrete algorithm) for learning finite-state controllers for POMDPs. The method is  inspired by previous methods from the verification literature. Experimental analysis is appropriate and shows improvements, and the paper is overall well written, though the material is not completely straightforward for the general UAI audience without a background in verification or FSC synthesis.

3 of 4 reviewers argue in favor of acceptance (with confidences 3, 2 and 1), whereas 2uBd argues for a borderline reject with confidence 4. I think this reflects the main issue of the paper: to an expert audience (the verification community), the paper is perhaps a bit incremental and lacks come comparison against SOTA algorithms in that community. However, the verification community seems less interested in finding local optima in POMDPs. On the other hand, approximately solving POMDPs is very interesting to the UAI community, but many members of that community lack a deep background in verification. Overall I think that the work is interesting for the wider UAI community---importantly, it seems is written well enough to allow non-expert readers to understand, follow, and jugde the work. While I think that the criticism raised by 2uBd is valid (and I would be curious to hear their verdict after the rebuttal, but they did not respond unfortunately), I am leaning towards suggesting acceptance at UAI. I think the paper's merits outweigh the criticism overall.

Details: the paper seems well written (though not all parts are straightforward to follow with a typical ML background), and the work could potentially be impactful to a larger part of the UAI community (interested in approximately solving POMDPs), not least by doing a good job in introducing a framework that is known in the verification community to UAI. While the approach is related to previous work in verification it still has sufficient originality.

Pro:
* Important topic, and non-standard approach in the UAI community (though less novel/original for the verification community)
* Paper is written well enough for readers not familiar with verification to follow
* Experiments and empirical results are good
* Most major criticism has been well addressed during the rebuttal (ACs interpretation, since all but one reviewer have not responded)

Cons:
* Limited novelty from a verification viewpoint
* Limited comparison against competitor approaches well known in the verification literature